

# Model calibration using ESEm v1.0.0 – an open, scalable Earth System Emulator

Duncan Watson-Parris[1], Andrew Williams[1], Lucia Deaconu[1], Philip Stier[1]

[1]Atmospheric, Oceanic and Planetary Physics, Department of Physics, University of Oxford, Oxford, UK

*Correspondence to*: Duncan Watson-Parris (duncan.watson-parris@physics.ox.uk)

**Abstract.** Large computer models are ubiquitous in the earth sciences. These models often have tens or hundreds of tuneable parameters and can take thousands of core-hours to run to completion while generating terabytes of output. It is becoming common practice to develop emulators as fast approximations, or surrogates, of these models in order to explore the relationships between these inputs
and outputs, understand uncertainties and generate large ensembles datasets. While the purpose of these surrogates may differ, their development is often very similar. Here we introduce ESEm: an open-source tool providing a general workflow for emulating and validating a wide variety of models and outputs. It includes efficient routines for sampling these emulators for the purpose of uncertainty quantification and model calibration. It is built on well-established, high-performance libraries to ensure
robustness, extensibility and scalability. We demonstrate the flexibility of ESEm through three case-studies using ESEm to reduce parametric uncertainty in a general circulation model, explore precipitation sensitivity in a cloud resolving model and scenario uncertainty in the CMIP6 multi-model ensemble.

## 1 Introduction

Computer models are crucial tools for their diagnostic and predictive power and are applied to every aspect of the earth sciences. These models have tended to increase in complexity to match the increasing availability of computational resources and are now routinely run on large supercomputers producing terabytes of output at a time. While this added complexity can bring new insights and improved accuracy, sometimes it can be useful to run fast approximations of these models, often
referred to as surrogates (Sacks et al., 1989). These surrogates have been used for many years to allow efficient exploration of the sensitivity of model output to its inputs (Lee et al., 2011b; Ryan et al., 2018), generation of large ensembles of model realisations (Holden et al., 2014, 2019; Williamson et al., 2013), and also model calibration (Holden et al., 2015a; Cleary et al., 2021) . Although relatively common, these workflows invariably use custom emulators and bespoke analysis routines, limiting their
reproducibility, and use by non-statisticians.

Here we introduce ESEm, a general tool for emulating earth systems models and a framework for using these emulators, with a focus on model calibration, broadly defined as finding model parameters which produce model outputs compatible with available observations. Unless otherwise stated, model parameters in this context refer to constant, scalar model inputs rather than e.g., boundary conditions.



This tool builds on the development of emulators for uncertainty quantification and constraint in the aerosol component of general circulation models (Regayre et al., 2018; Lee et al., 2011b; Johnson et al., 2018; Watson-Parris et al., 2020), but is applicable much more broadly as we will show.

Figure 1 shows a schematic of a typical model calibration workflow that ESEm enables, assuming a simple 'one shot' design for simplicity. Once the gridded model data has been generated it must be
collocated (resampled) on to the same temporal and spatial locations as the observational data which will be used to calibrate, in order to minimize sampling uncertainties (Schutgens et al., 2016a, b). The Community Intercomparison Suite (CIS; Watson-Parris et al., 2016) is an open-source Python library that makes this kind of operation very simple. The output is an Iris (Met Office, 2020b) Cube-like object, a representation of a Climate and Forecast (CF)-compliant NetCDF file, which includes all of
the necessary coordinate and metadata to ensure traceability and allow easy combination with other tools. ESEm uses the same representations throughout to allow easy input and output of the emulated datasets, plotting and validation and also allows chaining operations with other related tools such as Cartopy (Met Office, 2020a) and xarray (Hoyer and Hamman, 2016). Once the data has been read and collocated, it is split into training and validation (and optionally test) sets before performing emulation
over the training data using the ESEm interface. This emulator can then be validated and used for inference and calibration.

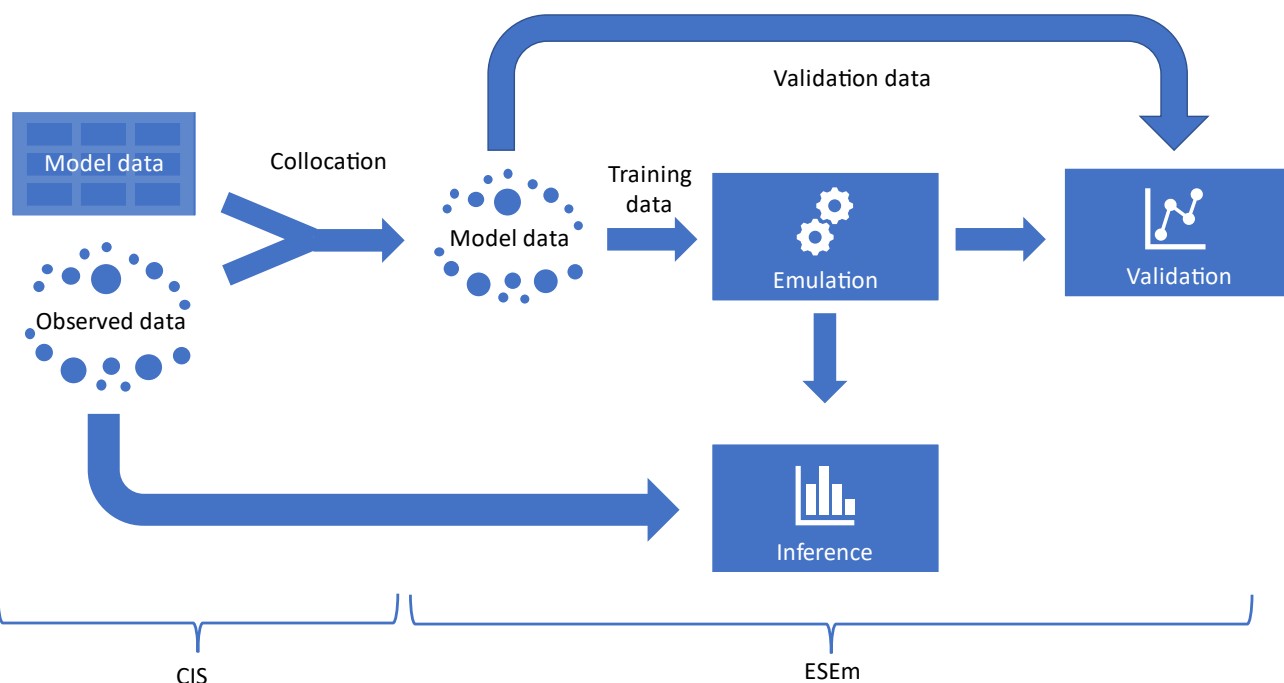

**Figure 1: A schematic of a typical workflow using CIS and ESEm to perform model emulation and calibration.**

Emulation is essentially a multi-dimensional regression problem and ESEm provides three main options
for performing these fits – Gaussian Processes (GPs), Convolutional Neural Networks (CNNs) and Random Forests (RFs). Based on a technique for estimating the location of gold in South Africa from




sparse mining information known as Krigging, and formalised by (Matheron, 1963), GPs have become a popular tool for non-parametric interpolation and an important tool within the field of supervised machine learning. Kennedy and O'Hagan (2001) first described the use of GPs for the calibration of
computer models which forms the basis of current approaches. GPs are particularly well suited to this task since they provide robust estimates and uncertainties to non-linear responses, even in cases with limited training data. Despite initial difficulties with their scalability as compared to e.g., Neural Networks, recent advances have allowed for deeper, more expressive (Damianou and Lawrence, 2012) GPs which can be trained on ever larger volumes of training data (Burt et al., 2019). Despite their
prevalent use in other areas of ML, CNNs and RFs have not been widely used in model emulation. Here we include both as examples of alternative approaches to demonstrate the flexible emulation interface as well as to motivate broader usage of the tool. For example, Section 5.1 shows the use of a RF emulator for exploring precipitation susceptibility in a cloud resolving model.

One common use of an emulator is to perform model calibration. By definition, any computer model
has a number of inputs and outputs. The model inputs can be high-dimensional boundary conditions or simple scalar parameters, and while large uncertainties can exist in the boundary conditions our focus here is on the latter. These input parameters can often be uncertain, either due to a lack of physical analogue, or lack of available data. Assuming that suitable observations of the model output are available, one may ask which values of the input parameters give the best output as measured against
the observations. This model 'tuning' is often done by hand leading to ambiguity and potentially sub-optimal configurations (Mauritsen et al., 2012). The difficulty in this task arises because, while the computer model is designed to calculate the output based on the inputs, the inverse process is normally not possible directly. In some cases, this inverse can be estimated and the process of generating an inverse of the model, known as inverse modelling, has a long history in hydrological modelling (e.g.
Hou and Rubin, 2005). The inverse of individual atmospheric model components can be determined using adjoint methods (Partridge et al., 2011; Karydis et al., 2012; Henze et al., 2007) but these require bespoke development and are not amenable to large multi-component models. Simple approaches can be used to determine chemical and aerosol emissions based on atmospheric composition but these implicitly assume that the relationship between emissions and atmospheric concentration is reasonably
well predicted by the model (Lee et al., 2011a). More generally, attempting to infer the best model inputs to match a given output is variously referred to as 'calibration', 'optimal parameter estimation' and 'constraining'. In many cases finding these optimum parameters requires very many evaluations of the model, which may not be feasible for large or complex models and so emulators are used as a surrogate. ESEm provides a number of options for performing this inference, from simple rejection
sampling to more complex Markov-Chain Monte-Carlo (MCMC) techniques.

Despite their increasing popularity, no general-purpose toolset exists for model emulation in the Earth sciences. Each project must create and validate their own emulators, with all of the associated data handling and visualisation code that necessarily accompanies them. Further, this code remains closed-source, discouraging replication and extension of the published work. In this paper we aim to not only
describe the ESEm tool, but also elucidate the general process of emulation with a number of distinct examples, including model calibration, in the hope of demonstrating its usefulness to the field. A description of the pedagogical example used to provide context for the framework description is





provided in Section 2, the emulation workflow and the two models included with ESEm is provided in Section 3, we then discuss the sampling of these emulators for inference in Section 4, before providing
two more specific example uses in Section 5 and some concluding remarks in Section 6.

## 2 Exemplar problem

While we endeavour to describe the technical implementation of ESEm in general terms, we will refer back to a specific example use-case throughout in order to aid clarity. This example case concerns the estimation of absorption aerosol optical depth (AAOD) due to anthropogenic black carbon (BC) which
is highly uncertain due to: limited observations and estimates of both pre-industrial and present-day biomass burning emissions; and large uncertainties in key microphysical processes and parameters in climate models (Bellouin et al., 2020).

Briefly, the model considered here is ECHAM6.3-HAM2.3 (Tegen et al., 2018; Neubauer et al., 2019) which calculates the distribution and evolution of both internally and externally mixed aerosol species
in the atmosphere and their effect on both radiation and cloud processes. We generate an ensemble of 39 model simulations for the year of 2017 over three uncertain input parameters: (1) a scaling of the emissions flux of BC by between 0.5 and 2 times the baseline emissions, (2) a scaling on the removal rate of BC through wet deposition (the main removal mechanism of BC) by between 1/3 and 3 times the baseline values, and (3) a scaling of the imaginary refractive index of BC (which determines its
absorptivity) between 0.2 and 0.8. The parameter sets are created using maximin latin-hypercube sampling where the scaling parameters (1 and 2) are sampled from log-uniform distributions, while the imaginary part of the refractive index is sampled from a normal distribution centered around 0.7. Unless otherwise stated five of the simulations are retained for testing while the rest are used for training the emulators. The model fields are emulated at their native resolution of approximately 1.8° longitude at
the equator (192 x 96 grid cells).

For simplicity, in this paper, we then compare the monthly mean model simulated aerosol absorption optical depth with observations of the same quantity in order to better constrain the global radiative effect of these perturbations. A full analysis including in-situ compositional and large-scale satellite observations, as well as an estimation of the effect of the constrained parameter space on estimates of
effective radiative forcing will be presented elsewhere.

Here we step through each of the emulation and inference procedures used to determine a reduced uncertainty in climate model parameters, and hence AAOD, by maximally utilising the available observations.

## 3 Emulation engines

Given the huge variety of geophysical models and their applications, and the broad (and rapidly expanding) variety of statistical models available to emulate them, ESEm uses an Object Oriented (OO) approach to provides a generic emulation interface. This interface is designed in such way as to





encourage additional model engines, either in the core package through pull-requests, or more informally as a community resource. The inputs include an Iris cube with the leading dimension representing the stack of training samples, and any other keyword arguments the emulator may require for training. Using either user specified or default options for the model hyper-parameters and optimisation techniques, the model is then easily fit to the training data and validated against the held-back validation data.

In this section we describe the inputs expected by the emulator and the three emulation engines provided by default in ESEm.

### 3.1 Input data preparation

In many circumstances the observations we would like to use to compare and calibrate our model against are provided on a very different spatial and temporal sampling than the model itself. Typically, a model might use a discretized representation of space-time, whereas observations are typically point-like measurements or retrievals. Naively comparing point observations with gridded model output can lead to large sampling biases (Schutgens et al., 2017). By collocating the models and observations, using CIS for example, we can minimise this error. An `ensemble_collocate` utility is provided in ESEm to use CIS to efficiently collocate multiple ensemble members on to the same observations. Other sources of observational-model error may still be present, and accounting for these will be discussed in Section 3.

In earth sciences these (resampled) model values are typically very large datasets with many millions of values. With sufficient computing power these can be emulated directly, however often there is a lot of redundancy in the data due to e.g., strong spatial and temporal correlations and this brute-force approach is wasteful. The use of summary statistics to reduce this volume while retaining most of the information content is a mature field (Prangle, 2015), and already widely used (albeit informally) in many cases. The summary statistic can be as simple as a global weighted average, or it could be an empirical orthogonal function (EOF) based approach (Ryan et al., 2018). Although some techniques for automatically finding such statistics are becoming available (Fearnhead and Prangle, 2012) this usually requires knowledge of the underlying data, and we leave this step for the user to perform using the standard tools available (e.g. Dawson, 2016).

Once the data has been resampled and summarised it should be split into training, validation and test sets. The training data is used to fit the models, while the validation portion of the data is used to measure their accuracy while exploring hyper-parameters. The test data is held back for final testing of the model. Typically, a 70:20:10 split is used. Excellent tools exist for preparing these splits, including for more advanced k-fold cross validation, and we include interfaces for such implementations in scikit-learn (Pedregosa et al., 2011), as well as routines for generating simple qualitative validation plots.

The input parameter space can also be reduced to enable more interpretable and robust emulation (also known as feature selection). ESEm provides a utility for filtering parameters based on the Bayesian (or Akaike) information content (BIC; Akaike, 1974) of the regression coefficients for a lasso least angle regression (LARS) model, using the scikit-learn implementation. This provides an objective estimate of





the importance of the different input parameters and allows removing any parameters which do not affect the output of interest.

**3.2 Gaussian Process engine**

Gaussian processes (GPs) are a popular choice for model emulation due to their simple formulation and
robust uncertainty estimates, particularly in cases of relatively small amounts of training data. Many excellent texts are available to describe their implementation and use (Rasmussen and Williams, 2005) and we only provide a short description here. Briefly, a GP is a stochastic process (a distribution of continuous functions) and can be thought of as an infinite dimensional normal distribution (hence the name). The statistical properties of the normal distributions and the tools of Bayesian inference allow
tractable estimation of the posterior distribution of functions given a set of training data. For a given mean function, a GP can be completely described by its second-order statistics and so the choice of covariance function (or kernel) can be thought of as a prior over the space of functions it can represent. Typical kernels include: constant; linear; radial basis function (RBF; or squared exponential); and Matérn 3/2 and 5/2 which are only once and twice differentiable respectively. Kernels can also be
designed to represent any aspect of the functions of interest such as non-stationarity or periodicity. This choice can often be informed by the physical setting and provides greater control and interpretability of the resulting model compared to e.g., Neural Networks. Fitting a GP involves an optimization of the remaining hyper-parameters, namely the kernel length-scale and smoothness.

A number of libraries are available which provide GP fitting, with varying degrees of maturity and
flexibility. By default, ESEm uses the open-source GPFlow (Matthews et al., 2017) library for GP based emulation. GPFlow builds on the heritage of the GPy library (GPy, 2012) but is based on the TensorFlow (Abadi et al., 2016) Machine Learning (ML) library with out-of-the-box support for the use of Graphical Processing Units (GPUs), which can considerably speed up the training of GPs. It also provides support for sparse and multi-output GPs. By default, ESEm uses a combination of linear, RBF
and polynomial kernels which are suitable for the smooth and continuous parameter response expected for the examples used in this paper and related problems. However, given the importance of the kernel for determining the form of the functions generated by the GP we have also included the ability for users to specify combinations of other common kernels. See e.g., (Duvenaud, 2011) for a clear description of some common kernels and their combinations, as well as work towards automated
methods for choosing them.

The framework provided by GPFlow also allows for multi-output GP regression and ESEm takes advantage of this to automatically provide regression over each of the output features provided in the training data. Figure 2 shows the emulated response from the ESEm generated GP emulation of absorption aerosol optical depth (AAOD) using a 'Bias + Linear' kernel for one specific set of the three
parameters outlined in Section 2 chosen from the test set (not shown during training). The emulator does an excellent job at reproducing the spatial structure of the AAOD for these parameters and exhibits errors less than an order of magnitude smaller than the predicted values, and significantly smaller than e.g., typical model and observational uncertainties.


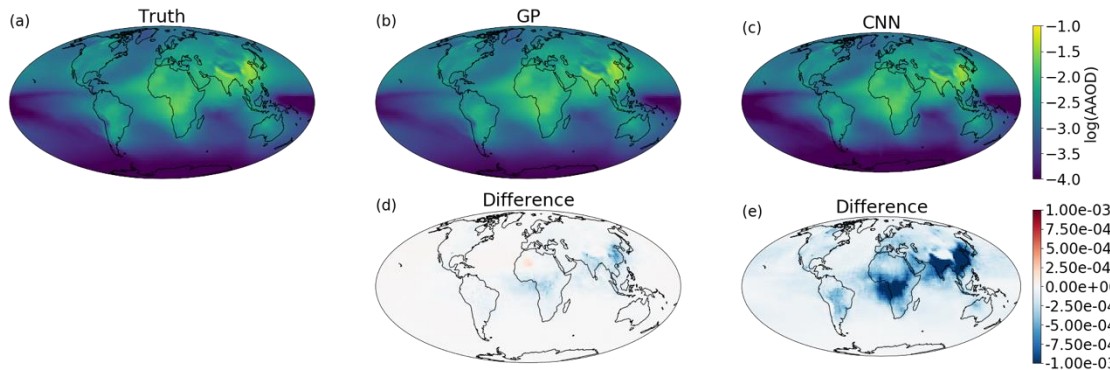

**Figure 2: Example emulation of absorption aerosol optical depth (AAOD) for a given set of three model parameters (broadly scaling emissions of black carbon, removal of black carbon, and the absorptivity of black carbon) as output by: (a) the full ECHAM-HAM aerosol-climate model; (b) a Gaussian process emulation; and (c) a convolutional neural network emulator, that were not trained on these parameters; as well as the differences between ECHAM-HAM and the emulators (d, e).**


### 3.3 Neural Network engine

Through the development of automatic differentiation and batch-gradient descent it has become possible to efficiently train very large (millions of parameters), deep (dozens of layers) neural networks, using large amounts (terabytes) of training data. The price of this scalability is the risk of overfitting,
and the lack of any information about the uncertainty of the outputs. However, both of these shortcomings can be addressed using a technique known as 'dropout' whereby individual weights are randomly set to zero and effectively 'dropped' from the network. During training this has the effect of forcing the network to learn redundant representations and reduce the risk of overfitting (Srivastava et al., 2014). More recently it was shown that applying the same technique during inference casts the NN
as approximating Bayesian inference in deep Gaussian processes and can provide a well calibrated uncertainty estimate on the outputs (Gal and Ghahramani, 2015). The convolutional layers within these networks also take into account spatial correlations which cannot currently be directly modelled by GPs (although dimension reduction in the input can have the same effect). The main drawback with a CNN based emulator is that they typically need a much larger amount of training data than GP based
emulators.

While fully connected neural networks have been used for many years, even in climate science (Knutti et al., 2006; Krasnopolsky et al., 2005), the recent surge in popularity has been powered by the increases in expressibility provided by deep, convolutional neural networks (CNNs) and the regularisation techniques which prevent these huge models from over-fitting the large amounts of
training data required to train them. Many excellent introductions can be found elsewhere but, briefly, a neural network consists of a network of nodes connecting (through a variety of architectures) the inputs to the target outputs via a series of weighted activation functions. The network architecture and activation functions are typically chosen a-priori and then the model weights are determined through a





combination of back-propagation and (batch) gradient descent until the outputs match (defined by a
given loss function) the provided training data. As previously discussed, the random dropping of nodes
(by setting the weights to zero), termed dropout, can provide estimates of the prediction uncertainty of
such networks. The computational efficiency of such networks and the rich variety of architectures
available have made them the tool of choice in many machine learning settings, and they are starting to
be used in climate sciences for emulation (Dagon et al., 2020), although the large amounts of training
data required have so far limited their use somewhat.

ESEm uses the Keras library (Chollet, 2015) with the TensorFlow backend to provide a flexible
interface for constructing and training CNN models and a simple, fairly shallow architecture is included
as an example. This default model takes the input parameters and passes them through an initial fully
connected layer before passing through two transpose convolutional layers which perform an inverse
convolution and act to 'spread-out' the parameter information spatially. The results of this default
model are shown in Figure 2c which shows the predicted AAOD from a specific set of three model
parameters. While the emulator clearly has some skill, and produces the large-scale structure of the
AAOD, the error compared to the full ECHAM-HAM output is larger than the GP emulator at around
10% of the absolute values. This is primarily due to the limited training data available in this example
(34 simulations). Also, this 'simple' network still contains nearly 1 million trainable parameters and so
an even simpler network would probably perform better given the linearity of the model response to
these parameters.

### 3.3 Random Forests

ESEm also provides the option for emulation with Random Forests using the open-source
implementation provided by scikit-learn. Random Forest estimators are comprised of an ensemble of
decision trees; each decision tree is a recursive binary partition over the training data and the predictions
are an average over the predictions of the decision trees (Breiman, 2001). As a result of this
architecture, Random Forests (along with other algorithms built on decision trees) have two main
attractions. Firstly, they require very little pre-processing of the inputs as the binary partitions are
invariant to monotonic rescaling of the training data. Secondly, and of particular importance for climate
problems, they are unable to extrapolate outside of their training data because the predictions are
averages over subsets of the training dataset. As a result of this, a Random Forest trained on output from
an idealized GCM was shown to automatically conserve water and energy (O'Gorman and Dwyer,
2018).

These features are of particular importance for problems involving the parameterization of sub-grid
processes in climate models (Beucler et al., 2021) and as such, although parameterization is not the
purpose of ESEm, we include a simple Random Forest implementation and hope to build on this in
future.





## 4 Calibration

Having trained a fast, robust emulator this can be used to calibrate our model against available observations. Generally, this problem involves estimating the model parameters which could give rise to, or best match, the available observations. More formally, we can define a model as a function $\mathcal{F}$ of input parameters $\theta$ and outputs $Y$: $\mathcal{F}(\theta) = Y$. Generally, both $\theta$ and $Y$ are high dimensional and may themselves be functions of space and time. Given a set of observations of $Y$, denoted $Y^0$, we would like

to calculate the inverse: $\mathcal{F}^{-1}(Y) = \theta$.

This inverse is unlikely to be well defined since many different combinations of parameters could feasibly result in a given output and so we take a probabilistic approach. In this framework we would like to know the posterior probability distribution of the input parameters: $p(\theta|Y^0)$. Using Bayes' theorem, we can write this as:

$$p(\theta|Y^0) = \frac{p(Y^0|\theta)p(\theta)}{p(Y^0)}$$ Eq. 1

Where the probability of an output given the input parameters, $p(Y^0|\theta)$, is referred to as the likelihood. While the model is capable of sampling this distribution, generally the full distribution is unknown and intractable, and we must approximate this likelihood.

Depending on the purpose of the calibration and assumptions about the form of $p(Y^0|Y)$, different techniques can be used. In order to determine a (conservative) estimate of the parametric uncertainty in

the model for example, we can use approximate Bayesian computation (ABC) to determine those parameters which are plausible given a set of observations. Alternatively, we may wish to know the optimal parameters to best match a set of observations and Markov-Chain Monte-Carlo based techniques might be more appropriate. Both of these sampling strategies are available in ESEm and we introduce each of them here.

### 4.1 Approximate Bayesian Computation

The simplest ABC approach seeks to approximate the likelihood using only samples from the simulator and a discrepancy function $\rho$:

$$p(\theta|Y^0) \propto p(Y^0|Y)p(Y|\theta)p(\theta) \approx \int \mathbb{I}(\rho(Y^0, Y) \leq \epsilon)\ p(Y|\theta)\ p(\theta)\ dY$$ Eq. 2

where the indicator function $\mathbb{I}(x) = \begin{cases} 1, & x\ is\ true \\ 0, & x\ is\ false \end{cases}$, and $\epsilon$ is a small discrepancy. This can then be

integrated numerically using e.g., Monte-Carlo sampling of $p(\theta)$. Any of those parameters for which

$\rho(Y^0, Y) \leq \epsilon$ are accepted and those which do not are rejected. As $\epsilon \to \infty$ therefore, all parameters are accepted and we recover $p(\theta)$. For $\epsilon = 0$, it can be shown that we generate samples from the posterior $p(\theta|Y^0)$ exactly.

In practice however the simulator proposals will never exactly match the observations and we must make a pragmatic choice for both $\rho$ and $\epsilon$. ESEm includes an implementation of the 'implausibility





metric' (Williamson et al., 2013; Craig et al., 1996; Vernon et al., 2010)(Williamson et al., 2013; Craig et al., 1996; Vernon et al., 2010) which defines the discrepancy in terms of the standardized Cartesian distance:

$$\rho(Y^0, Y(\theta)) = \frac{|Y^0 - Y(\theta)|}{\sqrt{\sigma_E^2 + \sigma_Y^2 + \sigma_R^2 + \sigma_S^2}} = \rho(Y^0, \theta) \qquad \text{Eq. 3}$$

where the total standard deviation is taken to be the squared sum of the emulator variance ($\sigma_E^2$) and the uncertainty in the observations ($\sigma_Y^2$) and due to representation ($\sigma_R^2$) and structural model uncertainties

($\sigma_S^2$). As described above, the representation uncertainty represents the degree to which observations at a particular time and location can be expected to match the (typically aggregate) model output (Schutgens et al., 2016a, b). While reasonable approximates can often be made of this and the observational uncertainties, the model structural uncertainties are typically unknown. In some cases, a multi-model ensemble may be available which can provide an indication of the structural uncertainties

for particular observables (Sexton et al., 1995), but these are likely to underestimate true structural uncertainties as models typically share many key processes and assumptions (Knutti et al., 2013). Indeed, one benefit of a comprehensive analysis of the parametric uncertainty of a model is that this structural uncertainty can be explored and determined (Williamson et al., 2015).

Framed in this way, $\epsilon$, can be thought of as representing the number of standard deviations the

(emulated) model value is from the observations. While this can be treated as a free parameter and may be specified in ESEm, it is common to choose $\epsilon = 3$ since it can be shown that for unimodal distributions values of $3\sigma$ correspond to a greater than 95% confidence bound (Vysochanskij and Petunin, 1980).

This approach is closely related to the approach of 'history matching' (Williamson et al., 2013) and can

be shown to be identical in the case of fixed $\epsilon$ and uniform priors (Holden et al., 2015b). The key difference being that history matching may result in an empty posterior distribution, that is, it may find *no* plausible model configurations which match the observations. With ABC on the other hand the epsilon is typically treated as a hyper-parameter which can be tuned in order to return a suitably large number of posterior samples. Both $\epsilon$ and the prior distributions can be specified in ESEm and it can thus

be used to perform either analysis. The speed at which samples can typically be generated from the emulator means we can keep $\epsilon$ fixed as in history matching and generate as many samples as is required to estimate the posterior distribution.

When multiple ($\mathcal{N}$) observations are used (as is often the case) $\rho$ can be written as a vector of implausibilities, $\rho(Y_i^O, \theta)$ or simply $\rho_i(\theta)$, and a modified method of rejection or acceptance must be

used. A simple choice is to require $\rho_i < \epsilon \; \forall \; i \; \in \mathcal{N}$, however this can become restrictive for large $\mathcal{N}$ due to the curse of dimensionality. The first step should be to reduce $\mathcal{N}$ through the use of summary statistics as described above. An alternative is to introduce a tolerance ($T$) such that only some proportion of $\rho_i$ need to be smaller than $\epsilon$: $\sum_{i=0}^{\mathcal{N}} H(\rho_i - \epsilon) < T$, where $H$ is the Heaviside function (Johnson et al. 2019), although this is a somewhat unsatisfactory approach that can hide potential

structural uncertainties. On the other hand, choosing $T = 0$ as a first approximation and then identifying





any particular observations which generate a very large implausibility provides a mechanism for identifying potential structural (or observational) errors. These can then be removed and noted for further investigation.

In order to illustrate this approach, we apply AERONET (AErosol RObotic NETwork) observations of
AAOD to the problem of constraining ECHAM-HAM model parameters as described in Section 2. The AERONET sun-photometers directly measure solar irradiances at the surface in clear-sky conditions, and by performing almucantar sky scans are able to estimate the single scattering albedo, and hence AAOD, of the aerosol in its vicinity (Dubovik and King, 2000; Holben et al., 1998). Daily average observations are taken from all available stations for 2017 and collocated with monthly model outputs
using linear interpolation. Figure 3 shows the posterior distribution for the parameters described in Section 2 if uniform priors are assumed and a Gaussian process emulator is calibrated with these observations. Lower values of both the imaginary part of the refractive index (IRI500) and the emissions scaling parameter (BCnumber) are shown to be more compatible with the observations than higher values, while the rate of wet deposition (Wetdep) is less constrained. Hence, higher values of
IRI500 and BCnumber can be ruled out as implausible given these observations (within the assumptions of our prior, GP model choices, and observational and structural model uncertainties).





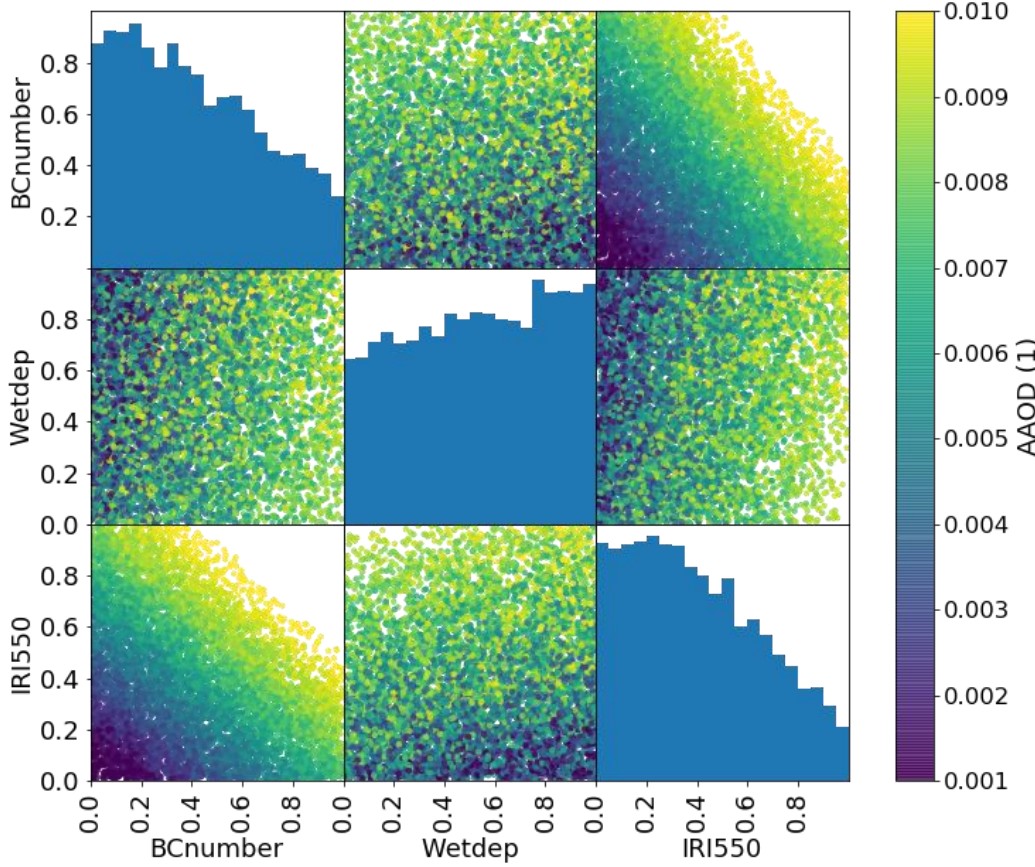

**Figure 3: The posterior distribution of parameters representing the plausible space of parameters for the example perturbed parameter ensemble experiment having been calibrated with a GP against observed absorbing aerosol optical depth measurements from AERONET. The diagonal histograms represent marginal distributions of each parameter while the off-diagonal scatter plots represent samples from the joint distributions. The colour represents the (average) emulated AAOD for each parameter combination.**

The matrix of implausibilities, $\rho_i(\theta)$, can also provide very useful information regarding the information content of each observation with respect to the various parameter combinations. Observations with narrow distributions of small implausibility provide little constraint value, whereas observations with a broad implausibility provide useful constraints on the parameters of interest. Observations with narrow distributions of high implausibility are useful indications of previously unknown structural uncertainties in the model.





### 4.2 Markov chain Monte-Carlo (MCMC)

The ABC method described above is simple and powerful, but somewhat inefficient as it repeatedly samples from the same prior. In reality each rejection or acceptance of a set of parameters provides us with extra information about the 'true' form of $p(\theta|Y^0)$ so that the sampler could spend more time in plausible regions of the parameter space. This can then allow us to use smaller values of $\epsilon$ and hence find better approximations of $p(\theta|Y^0)$.

Given the joint probability distribution described by Eq. 2 and an initial choice of parameters $\theta'$ and (emulated) output $Y'$, the acceptance probability $r$ of a new set of parameters ($\theta$) is given by:

$$r = \frac{p(Y^0|Y')p(\theta'|\theta)p(\theta')}{p(Y^0|Y)p(\theta|\theta')p(\theta)} \qquad \text{Eq. 4}$$

In the default implementation of MCMC calibration ESEm uses the TensorFlow-probability implementation of Hamiltonian Monte-Carlo (HMC) (Neal, 2011) which uses the gradient information automatically calculated by TensorFlow to inform the proposed new parameters $\theta$. For simplicity, we
assume that the proposal distribution is symmetric: $p(\theta'|\theta) = p(\theta|\theta')$, which is implemented as a zero log-acceptance correction in the initialisation of the TensorFlow target distribution. The target log probability provided to the TensorFlow HMC algorithm is then:

$$log(r) = log(p(Y^0|Y')) + log(p(\theta')) - log(p(Y^0|Y)) - log(p(\theta)) \qquad \text{Eq. 5}$$

Note, that for this implementation the distance metric $\rho$ must be cast as a probability distribution with values [0, 1]. We therefore assume that this discrepancy can be approximated as a normal distribution
centred about zero, with standard deviation equal to the sum of the squares of the variances as described in Eq. 3:

$$p(Y^0|Y) \approx \frac{1}{\sigma_t\sqrt{2\pi}}e^{-\frac{1}{2}\left(\frac{Y^0-Y}{\sigma_t}\right)^2}, \qquad \sigma_t = \sqrt{\sigma_E^2 + \sigma_Y^2 + \sigma_R^2 + \sigma_S^2} \qquad \text{Eq. 5}$$

The implementation will then return the requested number of accepted samples as well as reporting the acceptance rate, which provides a useful metric for tuning the algorithm. It should be noted that MCMC algorithms can be sensitive to a number of key parameters, including the number of burn-in steps used
(and discarded) before sampling occurs and the step size. Each of these can be controlled via keyword arguments to the sampler.

This approach can provide much more efficient sampling of the emulator and provide improved parameter estimates, especially when used with informative priors which can guide the sampler.

### 4.3 Extensions

While ABC and MCMC form the backbone of many parameter estimation techniques, there has been a large amount of research on improved techniques, particularly for complex simulators with high-dimensional outputs. See (Cranmer et al., 2020) for an excellent recent review of the state-of-the art techniques, including efforts to emulate the likelihood directly, utilising the 'likelihood ratio trick', and





even including information from the simulator itself (Brehmer et al., 2020). The sampling interface for
ESEm has been designed to decouple the emulation technique from the sampler and enable easy
implementation of additional samplers as required.

**5 Other use cases**

In order to demonstrate the generality of ESEm for performing emulation and/or inference over a
variety of earth science datasets here we introduce two further examples.

**5.1 Cloud-resolving model sensitivity**

In this example, we use an ensemble of large-domain simulations of realistic shallow cloud fields to
explore the sensitivity of shallow precipitation to local changes in the environment. The simulation data
we use for training the emulator is taken from a recent study (Dagan and Stier, 2020) which performed
ensemble daily simulations for one month-long period during December 2013 over the ocean to the East
of Barbados, sampling the variability associated with shallow convection. Each day of the month
consisted of two runs, both forced by realistic boundary conditions taken from reanalysis, but with
different cloud droplet number concentrations (CDNC) to represent clean and polluted conditions. The
altered CDNC was found to have little impact on the precipitation rate in the simulations, and so we
simply treat the CDNC change as a perturbation to the initial conditions and combine the two CDNC
runs from each day together to increase the amount of data available for training the emulator. At hourly
resolution, this provides 1488 data points.

However, given that precipitation is strongly tied to the local cloud regime, not fully controlling for
cloud regime can introduce spurious correlations when training the emulator. As such we also filter out
all hours which are not associated with shallow convective clouds. To do this, we consider domain-
mean vertical profiles of total cloud water content (liquid + ice), $q_t$, and filter out all hours where the
vertical sum of $q_t$ below 600hPa exceeds $10^{-6}$ kg/kg. This condition allows us to filter out hours
associated with the onset and development of deep convection in the domain, as well as masking out
hours with high cirrus layers or hours dominated by transient mesoscale convective activity which is
advected in by the boundary conditions. After this, we are left with 850 hourly data points which meet
our criteria and can be used to train the emulator.

As our predictors we choose five representative cloud controlling factors from the literature (Scott et al.,
2020), namely, in-cloud liquid water path (LWP), geopotential height at 700hPa ($z_{700}$), estimated
inversion strength (EIS), sea-surface temperature (SST) and the vertical pressure velocity at 700hPa
($w_{700}$). All quantities are domain-mean features and the LWP is a column average.

We then develop a regression model to predict shallow precipitation as a function of these five domain-
mean features using the scikit-learn Random Forest implementation within ESEm. After validating the
model using Leave-One-Out cross-validation, we then retrain the model using the full dataset, and use
this model to predict the precipitation across a wide range of values environmental values.





Finally, for the purpose of plotting, we reduce the dimensionality of our final prediction by averaging
over all features excluding LWP and $z_{700}$, and then plot in LWP-$z_{700}$ space. This allows us to
effectively account for, or marginalise out, those other environmental factors and investigate the
sensitivity of precipitation to LWP for a given $z_{700}$, as shown in Figure 4. LWP and $z_{700}$ were chosen
for plotting purposes as they are mutually uncorrelated and so span the two-dimensional space
effectively.

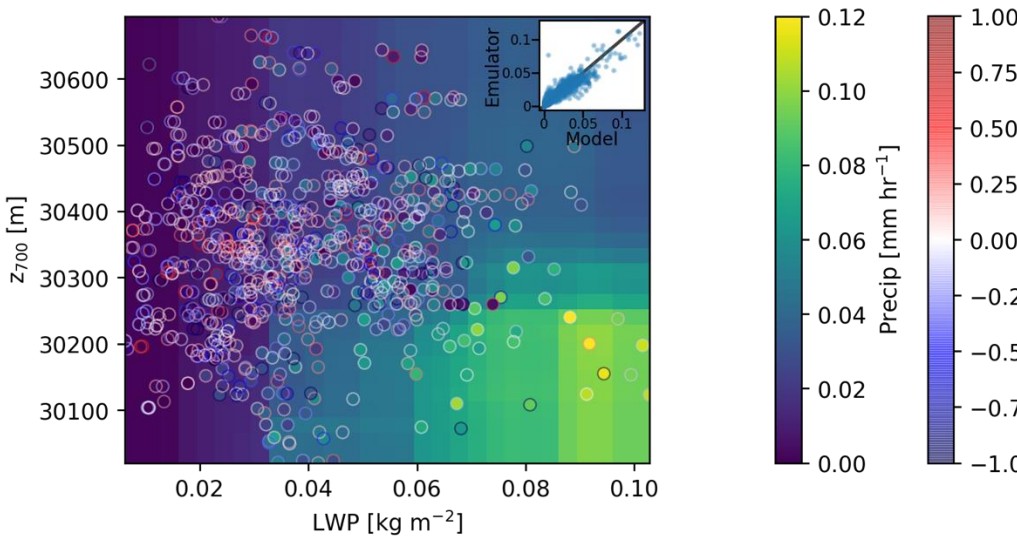


**Figure 4: The mean precipitation emulated by ESEm using a Random Forest model trained on the five environmental factors diagnosed from an ensemble of cloud resolving models as described in the text, plotted as a function of liquid water path and geopotential height at 700mb ($z_{700}$) by averaging over the remaining three dimensions corresponding to SST, EIS and $w_{700}$. The inset shows a validation plot of the emulated precipitation values against the model values using Leave-One-Out cross-validation.**
**The training points are shown as scatter points on the same scale. The scatter outlines represent the relative error between the emulator and the training data using the same data as in the inset.**

The inset in Figure 4 illustrates how the Random Forest regression model can capture most of the
variance in shallow precipitation from the cloud-resolving simulations, with an $R^2$ of 0.81 and a root
mean square error (RMSE) of 0.01 mm hr$^{-1}$. Additionally, the model captures basic physical features
such as the non-negativity of precipitation without requiring additional constraints. The coloured
surface in the main panel of Figure 4 shows the two-dimensional truncation of the model predictions
after averaging over all features except LWP and $z_{700}$, and shows that the model is behaving physically
by predicting an increase in precipitation at larger LWP and lower $z_{700}$.

While emulators have previously been used to investigate the behaviour of shallow cloud fields in high-
resolution models (e.g., using GPs, Glassmeier et al., 2019), this example demonstrates that Random
Forests are another promising approach, particularly due to their extrapolation properties.





## 5.2 Exploring CMIP6 scenario uncertainty

The 6[th] coupled model intercomparison project (CMIP6); (Eyring et al., 2016) coordinates a large
number of formal model intercomparison projects (MIPs), including ScenarioMIP (O'Neill et al., 2016)
which explored the climate response to a range of future emissions scenarios. While internal variability
and model uncertainty can dominate the uncertainties in future temperature responses to these future
emissions scenarios over the next 30-40 years, uncertainty in the scenarios themselves dominates the
total uncertainty by the end of the century (Hawkins and Sutton, 2009; Watson-Parris, 2021). Efficiently
exploring this uncertainty can be useful for policy makers to understand the full range of temperature
responses to different mitigation policies. While simple climate models are typically used for this
purpose (e.g., Smith et al., 2018; Geoffroy et al., 2013), statistical emulators can also be of use.

Here we provide a simple example of emulating the global mean surface temperature response to a
change in $CO_2$ concentration and aerosol loading. For these purposes we consider a change in aerosol
optical depth (AOD) and the cumulative emissions of $CO_2$ as compared to the start of the ScenarioMIP
simulations (averaged over 2015-2020). We use the global mean AOD and cumulative $CO_2$ at 2050 and
2100 for each model (11 models were used in this example: CanESM5, ACCESS-ESM1-5, ACCESS-
CM2, MPI-ESM1-2-HR, MIROC-ES2L, HadGEM3-GC31-LL, UKESM1-0-LL, MPI-ESM1-2-LR,
CESM2, CESM2-WACCM and NorESM2-LM) across the five main scenarios (SSP119, SSP126,
SSP245, SSP370, SSP585 and SSP434). The mean was taken over model submissions for which
multiple ensemble members were available to reduce model internal variability. As shown in Fig. 5, a
simple Gaussian Process regression model is able to fit the resulting temperature change well across the
range of training data. We can see that the emulator uncertainty increases away from the CMIP6 model
values as expected and largely reflects the inter-model spread within the range of scenarios explored
here.

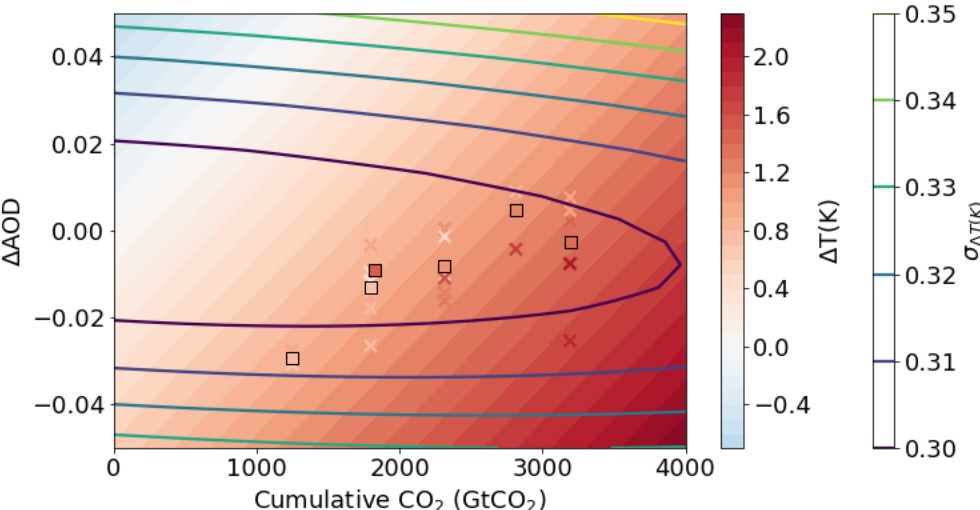






**Figure 5: Global mean surface temperature response to a change in aerosol optical depth (AOD) or cumulative atmospheric CO₂ concentration relative to the 2015-2020 average as emulated by ESEm using Gaussian process regression trained on CMIP6 ScenarioMIP outputs (shown as scatter points, the multi-model mean for each scenario are shown as square points). The contour lines represent the 1σ uncertainty in the emulator values, in Kelvin.**

Using a MCMC sampler, we are able to generate a joint probability distribution for the required change in AOD and CO₂ in order to meet 2.0℃ temperature rises since pre-industrial times as shown in Fig. 6 (assuming the present-day simulations start at +0.8℃ for simplicity). The effect of a decrease in (cooling) aerosol on the remaining carbon budget for a given temperature target is clear. It should be noted though that the short lifetime of aerosol means that while aerosol emissions can affect the year of
crossing a certain temperature threshold, stabilising at that temperature requires net-zero emissions of CO₂ regardless of the aerosol.

While more physically interpretable emulators are appropriate for such important estimates, the advantage these statistical emulators have over e.g., simple impulse response models is the ability to generalise to high-dimensional outputs, such as those shown in Fig. 2 (and e.g., Mansfield et al., 2020).
They can also account for the full complexity of Earth System Models and the many processes they represent. This is straightforward to achieve with ESEm.

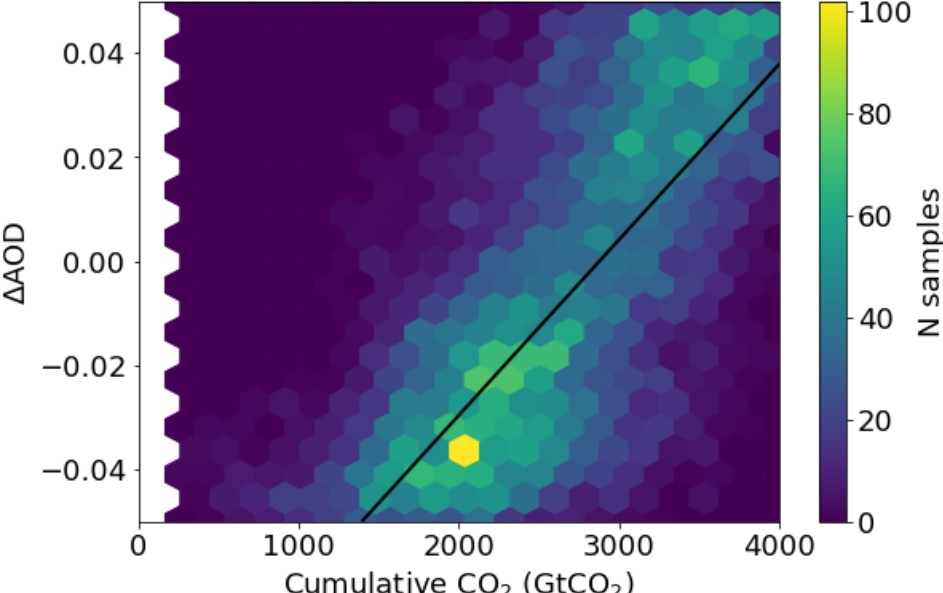

**Figure 6: The joint probability distribution for a change in aerosol optical depth (AOD) or cumulative atmospheric CO₂**
**concentration relative to the 2015-2020 average compatible with a change of 1.2K global mean surface temperature (approximately 2℃ above pre-industrial temperatures) as sampled from a Gaussian process emulator using MCMC accounting for emulator uncertainties. The solid black line corresponds to a change of 1.2K by interpolating the emulator surface shown in Figure 5.**



## 6 Conclusions

We present ESEm – a Python library for easily emulating and calibrating earth system models. Combined with the popular geospatial libraries Iris and CIS, ESEm makes reading, collocating and emulating a variety of model and earth system data straightforward. The package includes Gaussian process, Neural Network and Random Forest emulation engines and a minimal, clearly defined interface allows simple extension, as well as tools for validating these emulators. ESEm also includes two
popular techniques for calibration (or inference), optimised using TensorFlow to enable efficient sampling of the emulators. By building on fast and robust libraries in a modular way we hope to provide a framework for a variety of common workflows.

We have demonstrated the use of ESEm for model parameter constraint and optimal estimation with a simple perturbed parameter ensemble example. We have also shown how ESEm can be used to fit high-
dimensional response surfaces over an ensemble of cloud-resolving model simulations in order to determine the sensitivity of precipitation to environmental parameters in these simulations. Such approaches can also be useful in marginalising over, potentially confounding, variables in observational data. Finally, we presented the use of ESEm for the emulation of the multi-model CMIP6 ensemble in order to explore the global mean temperature response to changes in aerosol loading and $CO_2$
concentration in-between the handful of prescribed scenarios available in ScenarioMIP.

There are many opportunities to build on this framework and introduce the latest inference techniques (Brehmer et al., 2020), as well as bringing this setting of parameter estimation closer to the large body of work in data assimilation. While this has historically focussed on improving estimates of time-varying boundary conditions (the model 'state'), recent work is exploring using these approaches to
concurrently estimate constant model parameters (Brajard et al., 2020) We hope this tool will provide a useful framework in which to explore such ideas.

We strive to ensure reliability in the library through the use of automated unit-tests and coverage metrics. We also provide comprehensive documentation and a number of example notebooks to ensure useability and accessibility. Through the use of a number of worked examples we hope also to have
shed some light on this, at times, seemingly mysterious sub-field.

*Health warning:* While every effort has been made to make this tool easy to use and generally applicable, the example models provided make many implicit (and explicit) assumptions about the functional form and statistical properties of the data being modelled. Like any tool, the ESEm framework can be misused. Users should familiarise themselves with the models being used and consult
the many excellent textbooks on this subject if in any doubt as to their appropriateness for the task at hand.

## Author contributions

DWP designed the package and led its development. AD contributed the precipitation example and RF module. LD provided the AAOD example and dataset. PS provided supervision and funding
acquisition. DWP prepared the manuscript with contributions from all co-authors.



**Acknowledgements**

This work has evolved through numerous projects in collaboration with Ken Carslaw, Lindsay Lee, Leighton Regayre and Jill Johnson and we thank them for sharing their insights and R scripts from which this package is inspired. Those previous collaborations were funded by Natural Environment

Research Council (NERC) grants NE/G006148/1 (AEROS); NE/J024252/1 (GASSP), and E/P013406/1 (A-CURE) which we gratefully acknowledge.

For this work specifically, DWP and PS acknowledge funding from NERC projects NE/P013406/1 (A-CURE) and NE/S005390/1 (ACRUISE) as well as from the European Union's Horizon 2020 research and innovation programme iMIRACLI under Marie Skłodowska-Curie grant agreement No 860100. PS

additionally acknowledges support from the ERC project RECAP and the FORCeS project under the European Union's Horizon 2020 research programme with grant agreements 724602 and 821205. AW acknowledges funding from the Natural Environment Research Council, Oxford DTP, Award NE/S007474/1. LD acknowledges funding from NERC project NE/P013406/1 (A-CURE).

The authors also gratefully acknowledge useful discussions with Dino Sedjonovic and Daniel Partridge

as well as the support of Amazon Web Services through an AWS Machine Learning Research Award.

**Code availability**

The ESEm code, including that used to generate the plots in this paper is available here: https://doi.org/10.5281/zenodo.5196632.

**Data availability**

The BC PPE data is available here: https://zenodo.org/record/3856645. The ensemble CRM data is available here: https://zenodo.org/record/3785603. The raw CMIP6 data used here is available through the Earth System Grid Federation and can be accessed through different international nodes e.g.: https://esgf-index1.ceda.ac.uk/search/cmip6-ceda/. The derived dataset is available in the ESEm repository: https://doi.org/10.5281/zenodo.5196632

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
