# Peer review of "Model calibration using ESEm v1.1.0 – an open, scalable Earth System Emulator"

_Geoscientific Model Development, 2021_

## Referee Comment (RC2)

Referee's report on
*Model calibration using ESEm v1.0.0 -an open, scalable Earth System Emulator*
by Watson-Parris, Williams, Deaconu and Stier

Referee: Victoria Volodina

**General comments**

This paper presents ESEm, a Python library for emulating and calibrating Earth system models. Three widely used emulation techniques such as Gaussian process, Neural Network and Random Forest are provided as part of the tool. In addition, the ESEm also includes two calibration methods, specifically approximate Bayesian computation (ABC) and Markov-Chain Monte Carlo (MCMC). The authors highlight the importance of the proposed tool on three selected case studies. The goal of the authors is to propose an open-source and general tool for emulating and calibrating Earth system models.

Overall this is an interesting article. The authors managed to identify a gap in the process of tuning climate models and proposed a general tool to perform reproducible research. The authors managed to provide a range of examples to strengthen the case for using ESEm. However, I have a number of questions about the descriptions of emulation engines and calibration techniques outlined in the Specific comments section.

In addition, the authors claim that "no general-purpose toolset exists for model emulation in the Earth sciences", which is to the best of my knowledge is true. However, there has been extensive work done as part of the HIGH-TUNE project on tuning boundary-layer clouds parameterization with in-house developed High-Tune Explorer (htexplo) tool using GP emulators and multi-wave history matching [1, 2, 4]. The authors might find the description of the tools, the comparison with by-hand tuning and discussion about model discrepancy useful and constructive for their manuscript.

**Specific comments**

Further comments and questions are as follows.

1. In Section 3, it would be more instructive to provide a mathematical definition of the simulator and the respective surrogate model. In particular, I propose to move the mathematical formulation in Section 4, lines 277-279 to Section 3.

2. In Section 3.2, it would be helpful if you could provide a mathematical definition of the GP model. What form of the mean function are you using? Lines 167-170: you mentioned the reduction of input space using information criterion. Does it affect the number of terms in your mean function and/or kernel function? A helpful reference would be [1].

3. Line 204: for the demonstrator example, you used the 'Bias+Linear' kernel. Is there any connection between the Bias kernel and nugget term commonly specified for GP emulator? Is it a standard kernel choice?

4. In Section 3.3 Random Forests, I am left wondering whether Random Forest emulation would potentially be useful for approximating model responses with nonstationarity and discontinuity due to the binary partitions over the training data. Could you comment on this?

5. In Figure 2, there is a comparison between GP and CNN emulators. Could you also consider the Random Forests emulation strategy? If not, could you explain why?

6. In Section 4, line 278 you introduced a function $\mathcal{F}$ such that $\mathcal{F}(\theta) = Y$. However, in Equation (3), we observe that $Y$ is itself a function of $\theta$. Please revisit your notation.

7. In Section 4.1, I had some difficulties in following the description of Approximate Bayesian Computation (ABC). As I understand the authors are using ABC to approximate the likelihood $p(Y^0|\theta)$ in Equation (1) with samples from the simulator $Y$. In Equation (2), the authors defined

$$p(\theta|Y^0) \propto p(Y^0|Y)p(Y|\theta)p(\theta).$$

After comparing Equation (2) with Equation (1), we deduct that $p(Y^0|\theta) = p(Y^0|Y)p(Y|\theta)$, which cannot be right. Instead we require integration with respect to $Y$, i.e.

$$p(Y^0|\theta) = \int p(Y^0|Y)p(Y|\theta)dY$$

for this expression to be true. Therefore, Equation (2) becomes

$$p(\theta|Y^0) \propto \int p(Y^0|Y)p(Y|\theta)p(\theta)dY \approx \int I(\rho(Y^0,Y) \le \epsilon)p(Y|\theta)p(\theta)dY.$$

I am not an expert in ABC, but in Equation (2) we have an approximate sign ($\approx$), because you approximate probability function $p(Y^0|Y)$ with $I(\rho(Y^0,Y) \le \epsilon)$. Is it right? Perhaps it would be useful to provide readers with some ABC references.

8. I have difficulties in following Equation (3). In particular, the implausibility function commonly used in history matching is defined in terms of the first two moments, expectation and variance of the emulator. Instead in their implausibility computations, the authors use simulator output $Y(\theta)$ directly together with the emulator variance $\sigma_E^2$, which does not make sense. The implausibility function in Equation (3) should have the form

$$\rho(Y^0, Y(\theta)) = \frac{|Y^0 - \mu_E|}{\sqrt{\sigma_E^2 + \sigma_Y^2 + \sigma_R^2 + \sigma_S^2}},$$

where $\mu_E$ and $\sigma_E^2$ are the mean and variance of emulator respectively.

9. In Section 4.1, lines 333-343: the authors briefly discuss implausibilities for multiple observations. It would be useful to mention and reference multi-dimensional implausibility commonly used in history matching considered by Craig et al. (1996) and Vernon et al. (2010).

10. In Section 4.1, the authors provided an example to illustrate the ABC approach. I am curious to find out the percentage of input space that was retained, i.e. plausible space of parameters. This is a standard measure in history matching that could help to emphasise the importance of the proposed method.

11. In Section 4.2, lines 384-385: "...this discrepancy can be approximated as a normal distribution centred about zero... ". However, in Equation (5), $p(Y^0|Y)$ is a probability density function of a normal distribution centred around $Y$. Could you please clarify this point? Again, I am confused if the authors are using the simulator itself $Y$ instead of the emulator's mean and variance?

**Technical corrections**

1. Line 65: could you decipher the abbreviation ML in "prevalent use in other areas of ML"?

2. Line 115: Please provide the reference to maximin latin-hypercube sampling [3] in "The parameter sets are created using maximin latin-hypercube sampling..."

3. Figure 4 is hard to follow. It would be helpful to remove inset plot and produce two separate plots next to each other.

4. Figure 5: CMIP6 ScenarioMIP outputs and the multi-model mean for each scenario is very hard to detect from the provided plot.

**References**

[1] Fleur Couvreux, Frédéric Hourdin, Daniel Williamson, Romain Roehrig, Victoria Volodina, Najda Villefranque, Catherine Rio, Olivier Audouin, James Salter, Eric Bazile, et al. Process-based climate model development harnessing machine learning: I. A calibration tool for parameterization improvement. *Journal of Advances in Modeling Earth Systems*, 13(3):e2020MS002217, 2021.

[2] Frédéric Hourdin, Daniel Williamson, Catherine Rio, Fleur Couvreux, Romain Roehrig, Najda Villefranque, Ionela Musat, Laurent Fairhead, F Binta Diallo, and Victoria Volodina. Process-based climate model development harnessing machine learning: II. Model calibration from single column to global. *Journal of Advances in Modeling Earth Systems*, 13(6):e2020MS002225, 2021.

[3] Max D Morris and Toby J Mitchell. Exploratory designs for computational experiments. *Journal of statistical planning and inference*, 43(3):381–402, 1995.

[4] Najda Villefranque, Stéphane Blanco, Fleur Couvreux, Richard Fournier, Jacques Gautrais, Robin J Hogan, Frédéric Hourdin, Victoria Volodina, and Daniel Williamson. Process-Based Climate Model Development Harnessing Machine Learning: III. The Representation of Cumulus Geometry and Their 3D Radiative Effects. *Journal of Advances in Modeling Earth Systems*, 13(4):e2020MS002423, 2021.

---

## Author Comment (AC1)

"Model calibration using ESEm v1.0.0 – an open, scalable Earth System Emulator" by Duncan Watson-Parris et al.,

**Response to reviewers**

Firstly, we would like to thank the reviewers for their thorough and considered feedback on our paper which has helped to improve and clarify some key aspects.

We respond to each comment by the reviewers in turn, making comments in blue and highlighting particular changes to the revised manuscript in orange.

**Anonymous reviewer #1**

Watson-Parris et al. describe a new tool called ESEm for model emulation and calibration, and demonstrate its use through a variety of examples. ESEm seems like a valuable addition to the climate modeling research community, and appears to be very flexible to allow the user to adapt it to a variety of purposes. It is also open-source and easily available via GitHub. In general the manuscript is well written and organized. Some of the details could be clarified or expanded on, and I have highlighted these in the specific comments below. Once these are addressed, I recommend the paper for publication in GMD.

Specific comments:

Figure 1: What is represented by the arrow going from observations to model data (via collocation)? Does that represent resampling/regridding? It almost makes it seem like the observations are used for training, which I believe is not the case.

> Yes, this is supposed to represent the use of the observational locations in resampling the model data. We have tried to clarify this by adding "(resampling)" in the schematic and have added the following sentence to the Figure caption: "Note, only the locations of the observed data are used for resampling the model data."

Figure 1: Suggest adding "Calibration" somewhere to the flow chart, since it is a term mentioned in the caption and the text. Maybe in the same box as "Inference"?

> Yes, thank you. We have added "Calibration" to the inference box.

Line 54: What determined the three options for emulation? Were other machine learning and/or statistical models considered?

> These were determined based on their use in similar approaches in the literature as well as novel approaches to try and demonstrate the flexibility of the tool, as described on line 67.

Line 115: How were the parameter prior distributions determined in this case?

> These were determined by expert elicitation; we have now added a sentence to this effect: "These parameter ranges were determined by expert elicitation and designed to cover the broadest plausible range of uncertainty."

Line 118: Perhaps mixing terminology here: "five of the simulations are retained for testing" should that be validation instead of testing? Figure 1 and the accompanying description, as well as Line 138,

use "validation". Though later you discuss all three sets (e.g., Line 161ff), so some clarification on terminology is needed.

> Yes, this wasn't very clear. We do mean testing in this case as it is held back throughout the emulator development process, whereas validation data will be used for hyper-parameter tuning. We have added a signpost to the discussion you refer to, to aid the reader: "(see Section 3.1 for more details)." In that section we have added a brief discussion on the difference between validation and test data in the context of hyper-parameter optimisation as discussed below.

Line 119: Which model fields are emulated? AAOD only or others as well?

> No, only AAOD. We have clarified this sentence so that it now reads: "The model AAODs are emulated…"

Section 3: Is there anything built into ESEm to do hyper-parameter optimization/tuning? Or any suggested packages to automate that process? Suggest including that information somewhere in this section.

> Thank you, that is a good suggestion. We have added the following brief discussion:

> "Both scikit-learn and Keras (Chollet, 2015) include routines for automating the process of hyper-parameter optimization, with more advanced Bayesian optimization approaches available with the GPFlowOpt (Knudde et al., 2017) package. These all share many of the same dependencies as ESEm making installation very simple."

Line 150: Should that be Section 4?

> Yes, thank you. This has been corrected.

Line 159: Is resampling required for using the ESEm package?

> As mentioned in Line 154 large fields can be emulated directly but resampling is more important for calibration (as discussed in Line 339). We have added a comment to that effect in Line 156: "and can make calibration difficult (as discussed in Section 4)". Nevertheless, we have also added an "as required" caveat at the end of the paragraph to make it clear this isn't a strict requirement. The full paragraph now reads:

> *In earth sciences these (resampled) model values are typically very large datasets with many millions of values. With sufficient computing power these can be emulated directly, however often there is a lot of redundancy in the data due to e.g., strong spatial and temporal correlations and this brute-force approach is wasteful and can make calibration difficult (as discussed in Section 4). The use of summary statistics to reduce this volume while retaining most of the information content is a mature field (Prangle, 2015), and already widely used (albeit informally) in many cases. The summary statistic can be as simple as a global weighted average, or it could be an empirical orthogonal function (EOF) based approach (Ryan et al., 2018). Although some techniques for automatically finding such statistics are becoming available (Fearnhead and Prangle, 2012) this usually requires knowledge of the underlying data, and we leave this step for the user to perform using the standard tools available (e.g. Dawson, 2016) as required.*

Line 167ff: An alternative to this could be to apply feature importance tests to the trained emulators; it would be interesting to see how the results compare to filtering parameters before training via BIC/AIC.

> Yes, that would be very interesting. We have added a sentence to this effect at the end of the paragraph: "A complementary approach may be to apply feature importance tests to trained emulators to determine their sensitivity to particular input parameters."

Figure 2 caption: This part is unclear: "that were not trained on these parameters"? Suggest re-wording the caption.

> We have rephrased this as: "for parameter combinations that were not seen during training".

Line 219: Would Early Stopping also help with overfitting? As in, stopping training when the validation loss begins to increase beyond the training loss. How does that approach compare to dropout?

> Yes, it could help avoid over fitting. We have noted this in parentheses on Line 242:

> *While fully connected neural networks have been used for many years, even in climate science (Knutti et al., 2006; Krasnopolsky et al., 2005), the recent surge in popularity has been powered by the increases in expressibility provided by deep, convolutional neural networks (CNNs) and the regularisation techniques (such as early stopping) which prevent these huge models from over-fitting the large amounts of training data required to train them.*

Line 255: I wonder if a simple ANN would perform better. You could also reduce the number of layers/nodes to reduce the number of trainable parameters.

> Yes, it is quite possible that a well-designed ANN could perform well, although we wanted to highlight the possibility of using a CNN because it captures the spatial correlations that neither of the other approaches do. In conditions with more training data it could perform better than either of the other approaches.

Section 4 is nicely written to describe parameter calibration theory and how it is applied in practice for ESEm.

> Thank you, we tried to make this accessible!

Line 386: Equation 5 is labeled twice.

> Thank you, this is now corrected.

Figure 4: The scatter outline is difficult to distinguish by eye, perhaps the border width could be increased? I would also suggest adding a line to the text to help the reader interpret the meaning of this second colorbar (red/blue values). Or perhaps changing the colorbar label to something more intuitive, as described in the figure caption.

> Thank you for the suggestion. In addition to the changes suggested by Reviewer 2 (see below), we have increased the width of the border and added a few words describing this in the figure caption:

"…, with the scatter outlines showing the relative error between the emulator and the training data."

Figure 5: Suggest changing "scatter points" to "crosses" or something similar, to be clear which is which.

Yes, good point, this has been updated

**Victoria Volodina**

**General comments**

This paper presents ESEm, a Python library for emulating and calibrating Earth system models. Three widely used emulation techniques such as Gaussian process, Neural Network and Random Forest are provided as part of the tool. In addition, the ESEm also includes two calibration methods, specifically approximate Bayesian computation (ABC) and Markov-Chain Monte Carlo (MCMC). The authors highlight the importance of the proposed tool on three selected case studies. The goal of the authors is to propose an open-source and general tool for emulating and calibrating Earth system models.

Overall this is an interesting article. The authors managed to identify a gap in the process of tuning climate models and proposed a general tool to perform reproducible research. The authors managed to provide a range of examples to strengthen the case for using ESEm. However, I have a number of questions about the descriptions of emulation engines and calibration techniques outlined in the Specific comments section.

In addition, the authors claim that "no general-purpose toolset exists for model emulation in the Earth sciences", which is to the best of my knowledge is true. However, there has been extensive work done as part of the HIGH-TUNE project on tuning boundary-layer clouds parameterization with in-house developed High-Tune Explorer (htexplo) tool using GP emulators and multi-wave history matching [1, 2, 4]. The authors might find the description of the tools, the comparison with by-hand tuning and discussion about model discrepancy useful and constructive for their manuscript.

> Thank you for bringing these to our attention. We have added a citation to Couvreux et al. 2021 in our introduction.

**Specific comments**

Further comments and questions are as follows.

1. In Section 3, it would be more instructive to provide a mathematical definition of the simulator and the respective surrogate model. In particular, I propose to move the mathematical formulation in Section 4, lines 277-279 to Section 3.

> Thank you for the suggestion. We have tried to keep Section 3 focussed on the general approach of emulation and feel that since the emulators are not necessarily used for calibration and can be used for e.g., model exploration, this description fits best in Section 4.

2. In Section 3.2, it would be helpful if you could provide a mathematical definition of the GP model. What form of the mean function are you using? Lines 167-170: you mentioned the reduction of input space using information criterion. Does it affect the number of terms in your mean function and/or kernel function? A helpful reference would be [1].

> Given the generality of the GP model we don't feel that a mathematical description (which would just state that it is a function of the mean and covariance) would be particularly helpful. The choice of mean function and handling of active dimensions are very pertinent though and we have added reference to the mean function and two sentences discussing input dimensionality in the relevant paragraph:

> *A number of libraries are available which provide GP fitting, with varying degrees of maturity and flexibility. By default, ESEm uses the open-source GPFlow (Matthews et al., 2017) library for GP based emulation. GPFlow builds on the heritage of the GPy library (GPy,*

*2012) but is based on the TensorFlow (Abadi et al., 2016) machine learning library with out-of-the-box support for the use of Graphical Processing Units (GPUs), which can considerably speed up the training of GPs. It also provides support for sparse and multi-output GPs. By default, ESEm uses a* *zero mean and a* *combination of linear, RBF and polynomial kernels which are suitable for the smooth and continuous parameter response expected for the examples used in this paper and related problems. However, given the importance of the kernel for determining the form of the functions generated by the GP we have also included the ability for users to specify combinations of other common* *kernels and mean functions**. See e.g., (Duvenaud, 2011) for a clear description of some common kernels and their combinations, as well as work towards automated methods for choosing them.* *For stationary kernels, GPFlow automatically performs Automatic Relevance Determination (ARD) allowing lengthscales to be learnt independently for each input dimension. The user is also able to specify which dimensions should be active for each kernel in the case where the input dimension can be reduced (as discussed above).*

3. Line 204: for the demonstrator example, you used the `Bias+Linear' kernel. Is there any connection between the Bias kernel and nugget term commonly specified for GP emulator? Is it a standard kernel choice?

No, the Bias kernel is a simple Constant kernel that accounts for any offset in the data (in case the data hasn't been centred around zero). The nugget term can be accounted for separately, and learnt by the model if requested. We have made this clearer in the text:

*Figure 2 shows the emulated response from the ESEm generated GP emulation of absorption aerosol optical depth (AAOD) using a '**Constant* *+ Linear' kernel for one specific set of the three parameters outlined in Section 2 chosen from the test set (not shown during training).*

4. In Section 3.3 Random Forests, I am left wondering whether Random Forest emulation would potentially be useful for approximating model responses with non-stationarity and discontinuity due to the binary partitions over the training data. Could you comment on this?

Yes, that's a very good point, thank you. They would be naturally suited to these conditions, which have traditionally been difficult to emulate using GPs. We have added a short description of this possibility in the text:

"Finally, their construction as a combination of binary partitions lends itself to model responses that might be non-stationary or discontinuous."

5. In Figure 2, there is a comparison between GP and CNN emulators. Could you also consider the Random Forests emulation strategy? If not, could you explain why?

This was an oversight because the RF model was introduced after this plot was created. The emulated RF output is now also included in an updated Figure 2, thanks.

6. In Section 4, line 278 you introduced a function $\mathcal{F}$ such that $\mathcal{F}(\theta) = Y$. However, in Equation (3), we observe that $Y$ is itself a function of θ. Please revisit your notation.

Yes, this wasn't very clear and has now been rectified in Equation 3.

7. In Section 4.1, I had some difficulties in following the description of Approximate Bayesian Computation (ABC). As I understand the authors are using ABC to approximate the likelihood $p(Y|\theta)$ in Equation (1) with samples from the simulator $Y$. In Equation (2), the authors defined

$$p(\theta|Y^0) \propto p(Y^0|Y)p(Y|\theta)p(\theta)$$

After comparing Equation (2) with Equation (1), we deduct that $p(Y^0|\theta) = p(Y^0|Y)p(Y|\theta)$, which cannot be right. Instead, we require integration with respect to $Y$, i.e.

$$p(Y^0|\theta) = \int p(Y^0|Y)p(Y|\theta)dY$$

for this expression to be true. Therefore, Equation (2) becomes

$$p(\theta|Y^0) \propto \int p(Y^0|Y)p(Y|\theta)p(\theta)dY \approx \int \mathbb{I}(\rho(Y^0,Y) \leq \epsilon) \ p(Y|\theta) \ p(\theta) \ dY$$

I am not an expert in ABC, but in Equation (2) we have an approximate sign ($\approx$), because you approximate probability function $p(Y^0|Y)$ with $\mathbb{I}(\rho(Y^0,Y) \leq \epsilon)$. Is it right? Perhaps it would be useful to provide readers with some ABC references.

> Yes, thank you for catching this. We have updated Equation 2 to include the integral as suggested and included reference to the very useful 'Handbook of approximate Bayesian computation' (Sisson et al. 2019). This does not affect the code or results presented here.

8. I have difficulties in following Equation (3). In particular, the implausibility function commonly used in history matching is defined in terms of the first two moments, expectation and variance of the emulator. Instead in their implausibility computations, the authors use simulator output $Y(\theta)$ directly together with the emulator variance $\sigma_E^2$, which does not make sense. The implausibility function in Equation (3) should have the form

$$\rho\big(Y^0, Y(\theta)\big) = \frac{|Y^0 - \mu_E|}{\sqrt{\sigma_E^2 + \sigma_Y^2 + \sigma_R^2 + \sigma_S^2}}$$

where $\mu_E$ and $\sigma_E^2$ are the mean and variance of emulator respectively.

> Sorry for the confusion, yes, we use the emulator mean and variance as is usually done in history matching. We have updated Equation 3 to make this clear. Again, this does not affect the actual implenetation.

9. In Section 4.1, lines 333-343: the authors briefly discuss implausibilities for multiple observations. It would be useful to mention and reference multi-dimensional implausibility commonly used in history matching considered by Craig et al. (1996) and Vernon et al. (2010).

> Thank you for pointing this out. We have added a short description of these other common choices:
>
> *While the full multivariate implausibility can be estimated it requires careful consideration of the covariance structure (Vernon et al., 2010). An obvious choice is to require $\rho_i < \epsilon \ \forall \ i \in \mathcal{N}$, however this can become restrictive for large $\mathcal{N}$ due to the curse of dimensionality. The first step should be to reduce $\mathcal{N}$ through the use of summary statistics as described above. After that, the simplest solution is to require the maximum implausibility should be below our threshold: $\max_i\{\rho_i\} < \epsilon$ (e.g., Vernon et al., 2010).*

10. In Section 4.1, the authors provided an example to illustrate the ABC approach. I am curious to find out the percentage of input space that was retained, i.e. plausible space of parameters. This is a

standard measure in history matching that could help to emphasise the importance of the proposed method.

> Indeed, this is a key metric when performing history matching (or indeed MCMC sampling). We have added the following sentence to the relevant paragraph in Section 4.1, discussing Figure 3:

> Of the one million points sampled from this emulator, 729474 (73%) are retained as being compatible with the observations with $T = 0.1$.

11. In Section 4.2, lines 384-385: "...this discrepancy can be approximated as a normal distribution centred about zero... ". However, in Equation (5), $p(Y^0|Y)$ is a probability density function of a normal distribution centred around $Y$. Could you please clarify this point? Again, I am confused if the authors are using the simulator itself $Y$ instead of the emulator's mean and variance?

> Thank you, this was rather confusing. We have updated the description and Equation 6 to make it clear we are using the emulators mean and variance:

> We therefore assume that this discrepancy can be approximated as a normal distribution centred about the emulator mean ($\mu_E$), with standard deviation equal to the sum of the squares of the variances as described in Eq. 3:

$$p(Y^0|\mu_E) \approx \frac{1}{\sigma_t\sqrt{2\pi}} e^{-\frac{1}{2}\left(\frac{Y^0-\mu_E}{\sigma_t}\right)^2}, \qquad \sigma_t = \sqrt{\sigma_E^2 + \sigma_Y^2 + \sigma_R^2 + \sigma_S^2} \qquad \text{Eq. 6}$$

**Technical corrections**

1. Line 65: could you decipher the abbreviation ML in "prevalent use in other areas of ML"?

> This has been corrected, thank you.

2. Line 115: Please provide the reference to maximin latin-hypercube sampling [3] in "The parameter sets are created using maximin latin-hypercube sampling..."

> Good point, this has now been included.

3. Figure 4 is hard to follow. It would be helpful to remove inset plot and produce two separate plots next to each other.

> Agreed. This is now a separate panel in the same plot.

4. Figure 5: CMIP6 ScenarioMIP outputs and the multi-model mean for each scenario is very hard to detect from the provided plot.

> We had tried to make the comparison between the points and the background contours easy but it does make it hard to pick out the points. We have changed the model markers to be circles and added thick borders to all points to make them easier to see.

**References**

[1] Fleur Couvreux, Frederic Hourdin, Daniel Williamson, Romain Roehrig, Victoria Volodina, Najda Villefranque, Catherine Rio, Olivier Audouin, James Salter, Eric Bazile, et al. Process-based climate model development harnessing machine learning: I. A calibration tool for parameterization improvement. Journal of Advances in Modeling Earth Systems, 13(3):e2020MS002217, 2021.

[2] Frederic Hourdin, Daniel Williamson, Catherine Rio, Fleur Couvreux, Romain Roehrig, Najda Villefranque, Ionela Musat, Laurent Fairhead, F Binta Diallo, and Victoria Volodina. Process-based climate model development harnessing machine learning: II. Model calibration from single column to global. Journal of Advances in Modeling Earth Systems, 13(6):e2020MS002225, 2021.

[3] Max D Morris and Toby J Mitchell. Exploratory designs for computational experiments. Journal of statistical planning and inference, 43(3):381{402, 1995.

[4] Najda Villefranque, Stephane Blanco, Fleur Couvreux, Richard Fournier, Jacques Gautrais, Robin J Hogan, Frederic Hourdin, Victoria Volodina, and Daniel Williamson. Process-Based Climate Model Development Harnessing Machine Learning: III. The Representation of Cumulus Geometry and Their 3D Radiative Effects. Journal of Advances in Modeling Earth Systems, 13(4):e2020MS002423, 2021.